# Differing Nutrient Intake and Dietary Patterns According to the Presence of Hyper-Low-Density Lipoprotein Cholesterolemia or Hypertriglyceridemia

**DOI:** 10.3390/nu13093008

**Published:** 2021-08-28

**Authors:** Yu-Jin Kwon, Sujee Lee, Hye Sun Lee, Ji-Won Lee

**Affiliations:** 1Department of Family Medicine, Yongin Severance Hospital, Yonsei University College of Medicine, Yongin 16995, Korea; digda3@yuhs.ac; 2Department of Research Affairs, Biostatistics Collaboration Unit, Yonsei University College of Medicine, Seoul 06273, Korea; LEEVERDA@yuhs.ac; 3Department of Family Medicine, Gangnam Severance Hospital, Yonsei University College of Medicine, Seoul 06273, Korea

**Keywords:** low-density lipoprotein cholesterol, triglyceride, nutrition, Korean Health Eating Index

## Abstract

Dietary choices may have differing effects on low-density lipoprotein cholesterol or triglyceride levels. The aim of this study was to investigate daily nutrient intake and dietary patterns of individuals with hyper-low-density lipoprotein cholesterolemia (hLDL) and hypertriglyceridemia (hTG) in a large Korean population-based study using propensity score (PS) matching. This study used data from the Korea National Health and Nutrition Examination Survey. Propensity score values for the predicted probability of patients with hLDL or hTG were estimated using logistic regression analysis, with age, sex, body mass index, alcohol consumption, smoking status, physical activity status, hypertension, and diabetes. After PS matching, intake of carbohydrates (%) was significantly lower (*p* = 0.021), and intake of fats (%) and saturated fatty acids (%) was significantly higher in the hLDL group than in the non-hLDL group (*p* = 0.025 and *p* = 0.013, respectively). The percentage of individuals with a high score for the Korean Healthy Eating Index (KHEI) “whole grains” or “saturated fatty acids” components was higher in the non-hLDL group than in the hLDL group (*p* < 0.05 for both). Dietary sodium/potassium ratio was significantly higher in the hTG than in the non-hTG (*p* = 0.049). Our results suggest that individualized dietary information and counseling require consideration of a person’s specific lipid levels.

## 1. Introduction

Cardiovascular disease (CVD) is a major cause of morbidity and mortality globally and in Korea [1,2]. Dyslipidemia, characterized by elevated total cholesterol (TC), low-density lipoprotein cholesterol (LDL-C), and triglycerides (TG) levels and decreased high-density lipoprotein cholesterol (HDL-C), has been closely linked to the development of CVD and is a modifiable risk factor using lifestyle management [3]. In recent years, dyslipidemia has become a substantial disease burden in Korea. According to the Korean Society of Lipid and Atherosclerosis (KSoLA), the number of people diagnosed with dyslipidemia has abruptly increased nearly 8-fold, from 1.5 million in 2002 to 11.6 million in 2018. Approximately 40.5% of Korean adults >30 years had dyslipidemia in 2018 [4], which is comparable to the prevalence in other high-income countries, such as the United States and Japan [5,6]. This trend may reflect a shift in Koreans’ dietary habits toward more Western-type diets [7]. Promoting a healthy diet forms the cornerstone for treating dyslipidemia [8]. Typically, high consumption of fruits, vegetables, whole grains, fish rich in omega-3 polyunsaturated fatty acids (N-3 PUFA), and legumes is recommended, whereas saturated fatty acids (SFA), trans fatty acids, added sugars, and sodium (Na) are restricted [9,10]. In addition, certain dietary patterns, such as the Mediterranean diet and the Dietary Approaches to Stop Hypertension (DASH) diet, are recommended for their beneficial effects on dyslipidemia [11,12]. However, dietary choices influencing LDL-C or TG levels differ [8]. For example, consuming a low-fat, high-carbohydrate diet lowers LDL-C but raises TG, thereby failing to improve the overall blood lipid profile [13,14]. Developing specific tools for reducing hyper-LDL-cholesterolemia (hLDL) and hypertriglyceridemia (hTG) would therefore require separate clarification and strategies. Nevertheless, very few studies have examined how specific nutrients or dietary patterns differentially affect hLDL and hTG, especially in Asian populations.

Therefore, the aims of this study were to investigate daily nutrient intake and dietary habits of people with hLDL or hTG in a large Korean population-based study using propensity score (PS) matching and to compare the consumption of daily nutrients between these two dyslipidemia phenotypes.

## 2. Materials and Methods

### 2.1. Study Population

This study used secondary data obtained from the Korea National Health and Nutrition Examination Survey (KNHANES), a survey conducted by the Korea Centers for Disease Control and Prevention to assess the health and nutrition status of Koreans. The data are publicly available through the KNHANES website.

We used data from the 2016–2018 KNHANES (VII) survey. A total of 24,269 participants were included in KNHANES-VII. Exclusion criteria for this study were as follows: (1) age < 19 years (*n* = 4880), (2) missing serum LDL-C data (*n* = 1591), or (3) missing serum TG data (*n* = 3741). In total, 14,057 participants were included in this study. The study flow chart is shown in Figure 1. The Institutional Review Board of Yongin Severance Hospital approved the study protocol. This study was approved by the Institutional Review Board of Korea Centers for Disease Control and Prevention (IRB No: 2018–01-03-P-A). Informed consent was obtained from all subjects before KNHANES, in accordance with the Declaration of Helsinki.

### 2.2. Anthropometrics and Biochemical Variables

The health interview and health examination were conducted by trained staff members, including physicians, medical technicians, and health interviewers, at a mobile examination center. Body mass index (BMI) was calculated as weight (kg) divided by height (m) squared. Waist circumference (WC) was measured in the horizontal plane midway between the lowest rib and iliac crest. Blood pressure was measured two times with the patient in the sitting position. The mean of the two measurements was recorded. TC and TG levels were measured by an enzymatic method using the Hitachi Automatic Analyzer 7600–210 (Hitachi, Tokyo, Japan). LDL-C and HDL-C were measured by homogeneous enzymatic colorimetric methods using the same analyzer. In KNHANES-VII, LDL-C was measured directly when TG levels were ≥400 mg/dL [15]. When TG levels were <400 mg/dL, LDL-C was calculated using the Friedwald equation (LDL-C = TC-HDL-C-(TG/5)), which has been demonstrated to correlate with direct LDL-C measurements [16]. Lifestyle data, such as smoking, physical activity, and alcohol consumption, were reported using a self-administered questionnaire. Current smoker was defined as smoking cigarettes at present plus smoking ≥100 cigarettes during the person’s lifetime. Current alcohol drinker was defined as consumption of >1 cup of alcohol per week during the past 1 year. Physically active was defined as participating in ≥150 min of weekly moderate-intensity aerobic physical activity, ≥75 min of weekly vigorous-intensity aerobic physical activity, or an equivalent combination of moderate- and vigorous-intensity activity. The presence of hypertension was defined as systolic blood pressure (SBP) ≥ 160 mm Hg, diastolic blood pressure (DBP) ≥ 140 mm Hg, or anti-hypertensive medication use. The presence of diabetes was defined as a fasting glucose ≥126 mg/dL or anti-diabetes medication use.

### 2.3. Definition of Dyslipidemia

According to the National Cholesterol Education Program Adult Treatment Panel III criteria and KSoLA [17], hLDL was defined as a serum LDL-C ≥ 160 mg/dL or use of a cholesterol-lowering drug. hTG was defined as a serum TG level ≥ 200 mg/dL.

### 2.4. Assessment of Nutrition Intake and Dietary Patterns

The nutritional survey was a face-to-face interview conducted by skilled dietitians in the subjects’ homes. Food intake during the past 1 year was examined using the semi-quantitative food frequency questionnaire (FFQ), which has been validated for 112 food items [18]. Total energy and nutrient intake values were derived from this FFQ. Details regarding food items are summarized on the KNHANES website [15]. The survey staff completed an initial intensive, standardized course, and retraining courses are provided 5–6 times per year to reinforce the proper protocols and techniques [19]. Percentages of energy intake from carbohydrates or protein intake were calculated as follows: carbohydrate intake (g/day) × 4 kcal/total energy intake (kcal/day) × 100% and protein intake (g/day) × 4 kcal/total energy intake (kcal/day) × 100%. The percentages of energy intake from fats (total fats, SFA, monounsaturated fatty acids (MUFA), total polyunsaturated fatty acids (PUFA), N-3 PUFA, and omega-6 PUFA (N-6 PUFA)) were calculated as follows: fat intake (g/day) × 9 kcal/total energy intake (kcal/day) × 100%. Cholesterol intake was recorded in mg/day. Intake of Na and potassium (K) was recorded in g/day.

The Korean Healthy Eating Index (KHEI) was developed to assess adherence to national dietary guidelines and to provide a comprehensive evaluation of diet quality in healthy Korean adults [20]. KHEI contains 14 components: 8 items recommended for adequate food consumption (i.e., having breakfast; whole grains; fruits, excluding fruit juice; fruits, including fruit juice; total vegetables, including Kimchi and pickles; total vegetables, excluding Kimchi and pickles; meat, fish, eggs, legumes; and milk and dairy products); 6 items recommended for moderate food consumption (i.e., SFA, Na, empty-calorie foods (sweets, beverages), carbohydrates, total fats, and total energy). Adequacy components represent the food groups and dietary elements that are encouraged. For these components, higher scores reflect higher intakes as higher intakes are desirable. Moderation components represent the food groups and dietary elements for which there are recommended limits to consumption. For moderation components, higher scores reflect lower intakes as lower intakes are more desirable. Overall, a higher total KHEI score indicates a diet that aligns better with dietary recommendations. The maximum total score of all 14 components is 100 points, and each component has a maximum score of 5 or 10 points. Detailed information regarding KHEI is summarized in the previous study in Appendix A [21]. A higher KHEI score reflects a higher-quality, healthier diet. We categorized each component into three groups: low score, 0 ≤ score < 4, middle score: 4 ≤ score < 7, and high score: ≥7 points. For components with a maximum of 5 points, we doubled the points and categorized the component using the same three groups.

### 2.5. Statistical Analysis

Data were expressed as mean ± standard deviation (SD) for continuous variables or number (percentage) for categorical variables. To compare general characteristics between groups, an independent two-sample *t*-test was used for continuous variables, whereas a chi-squared test was used for categorical variables. PS values for the predicted probability of patients with hLDL or hTG were estimated using logistic regression analysis, with age, sex, BMI, alcohol consumption, smoking status, physical activity status, hypertension, and diabetes as confounding variables. Participants with hLDL were matched to non-hLDL participants in a 1:1 manner (Figure 1) using a nearest-neighbor matching method with a greedy algorithm. Participants with hTG were similarly 1:1 matched to non-hTG individuals. Participants with hLDL but not hTG (LDL-only group) were matched to those with hTG but not hLDL (hTG-only group) in the same manner.

Statistical analyses were conducted using SAS version 9.4 (SAS Institute, Cary, NC, USA) and R Statistical Package (Institute for Statistics and Mathematics, Vienna, Austria, ver 4.1.0, www.R-project.org). *p* < 0.05 was used as the significance level.

## 3. Results

### 3.1. Clinical and Dietary Characteristics According to LDL-C Level

The mean age and BMI of the 14,057 study participants were 52.0 ± 16.6 y and 23.9 ± 3.5 kg/m^2^. Table 1 shows the clinical characteristics of the study population according to LDL-C level. A total of 1312 participants (9.3%) had hLDL (hLDL group) and 12,745 (90.7%) did not have hLDL (non-hLDL group). In unmatched data, age, BMI, and WC were significantly higher, and the percentage of men was significantly lower in the hLDL group than in the non-hLDL group. Individuals with hLDL were less likely to be current smokers, current alcohol drinkers, and physically active. They also had significantly higher SBP, DBP, TC, LDL-C, HDL-C, C-reactive protein (CRP), and uric acid than people in the non-hLDL group.

After 1:1 PS matching according to age, sex, BMI, alcohol consumption, smoking, physical activity, hypertension, and diabetes, a total of 2966 individuals were matched based on LDL cholesterol levels. The 1483 matched sets were analyzed for differences in various metabolic variables (Table 1). The hLDL group had significantly higher SBP, DBP, glucose, TC, TG, LDL-C, HDL-C, and uric acid levels compared to the non-hLDL group.

Table 2 shows daily nutrient intake according to LDL-C levels. In unmatched data, there were no differences in total caloric intake or intake of carbohydrates (% and g), proteins (% and g), fats (% and g), SFA, MUFA, total PUFA, N-6 PUFA, fiber, or potassium between hLDL and non-hLDL groups. N-3 PUFA intake was significantly higher in the hLDL group than in the non-hLDL group (*p* = 0.022). Dietary Na and dietary Na/K ratio were lower in the hLDL group (*p* = 0.048 and *p* = 0.002, respectively). After 1:1 PS matching, total caloric intake was not different between hLDL and non-hLDL groups. Carbohydrate (%) intake was significantly lower (*p* = 0.021) and fat (%) and SFA intakes were significantly higher (*p* = 0.025 and *p* = 0.013, respectively) in the hLDL group than in the non-hLDL group.

The mean total KHEI score in the study population was 63.8 ± 13.5. The total KHEI score was not significantly different between hLDL and non-hLDL groups before and after PS matching (data not shown). When comparing the percentage of individuals in each KHEI category between hLDL and non-hLDL groups, the percentage of people with a high score for “whole grains” or “SFA” was significantly higher in the non-hLDL group than in the hLDL group after PS matching (*p* < 0.05 for both comparisons) (Figure 2A). There were no other differences between hLDL and non-hLDL groups.

### 3.2. Clinical and Dietary Characteristics According to TG Level

Table 3 shows the clinical characteristics of the study population according to TG level. Among the total study population, 2042 individuals (14.5%) had hTG (hTG group) and 12,015 (83.4%) did not have hTG (non-hTG group). In unmatched data, age, BMI, WC, and percentage of men were significantly higher in the hTG group than in the non-hTG group. Individuals in the hTG group were also more likely to be current smokers and current alcohol drinkers and less likely to be physically active. The hTG group also had significantly higher SBP, DBP, glucose, TC, TG, white blood cell (WBC) count, CRP, and uric acid than the non-hTG group. However, LDL-C and HDL-C levels were significantly lower in the hTG group than in the non-hTG group.

After 1:1 PS matching according to age, sex, BMI, alcohol consumption, smoking status, physical activity status, hypertension, and diabetes, a total of 3590 individuals were matched based on TG levels. The 1795 matched sets were analyzed for differences in various metabolic variables (Table 3). The hTG group had significantly higher SBP, DBP, WC, TC, TG, WBC count, and uric acid than the non-hTG group. LDL-C and HDL-C remained significantly lower in the hTG group than in the non-hTG group.

Table 4 shows daily nutrient intake according to TG levels. In unmatched data, there were significant differences in intake of all nutrients except protein (%), fat (g), N-3 PUFA, and cholesterol. Compared to the non-hTG group, the hTG group had significantly higher total caloric intake; intake of carbohydrates (% and g), fiber, Na, and K; and dietary Na/K ratio. Conversely, intake of fats (%), SFA, MUFA, total PUFA, and N-6 PUFA were significantly lower in the hTG group than in the non-hTG group. After 1:1 PS matching, there were no significant differences between hTG and non-hTG groups for all nutrient intake parameters except the dietary Na/K ratio. The Na/K ratio was significantly higher in the hTG group than in the non-hTG group (*p* = 0.049).

Total KHEI scores differed between hTG and non-hTG groups (62.3 ± 13.3 versus 64.3 ± 12.4 versus, respectively; *p* < 0.001). After PS matching, the total KHEI score remained significantly higher in the non-hTG group (64.4 ± 13.4 versus 62.9 ± 13.1; *p* = 0.045). As shown in Figure 2B, the percentage of individuals with a high score for “having breakfast” or “milk and dairy products” was significantly higher in the non-hTG group than in the hTG group after PS matching (*p* < 0.05 for both comparisons). There were no other significant differences between hTG and non-hTG groups.

### 3.3. Comparison of Nutrition Intake between hLDL-Only and hTG-Only Groups

Table 5 presents comparisons of nutrition intake between individuals in the hLDL-only and hTG-only groups. In unmatched data, the hLDL-only group had higher fat (%) and SFA intake (%) than the hTG-only group. The hTG-only group had a higher total caloric intake, PUFA/SFA ratio, (PUFA+MUFA)/SFA ratio, Na intake, and dietary Na/K ratio than the hLDL-only group. After 1:1 PS matching, the hLDL-only group had a higher intake of protein (%), fat (%), SFA, MUFA, total PUFA, N-3 PUFA, and N-6 PUFA. By contrast, the hTG-only group had a higher carbohydrate (% and g) intake and dietary Na/K ratio than the hLDL-only group.

## 4. Discussion

Using PS matching analysis, we found that individuals with hLDL had higher fat and SFA intake, lower carbohydrate intake, and a lower dietary Na/K ratio than individuals without hLDL. In addition, individuals with hTG had a higher dietary Na/K ratio than individuals without hLDL. When comparing individuals with only hLDL versus those with only hTG, the hLDL-only group had a higher intake of protein and fat, including various fatty acids, whereas the hTG-only group had a higher carbohydrate intake and a higher dietary Na/K ratio.

As dyslipidemia is strongly linked to mortality and morbidity from CVD [3], understanding dietary habits that may promote or prevent dyslipidemia is important for dyslipidemia management and CVD prevention [22]. Reducing LDL-C is a main focus of CVD management, and hTG is also an important consideration [23,24]. Further, hTG may be an especially important risk factor for CVD in Asian populations, whose carbohydrate consumption is higher but fat intake is much lower than that of Western populations [25,26]. A number of dietary habits have divergent effects on LDL-C and TG [8,27]. For example, dietary carbohydrate intake has unfavorable effects on hTG and HDL-C, but it is beneficial with regard to LDL-C [27]. Conversely, omega-3 intake from fish oil has no beneficial effects on LDL-C but lowers TG [28]. In a meta-analysis including 21trials, fish oil supplementation was associated with an increase in LDL-C levels [29]. Therefore, hLDL and hTG may require different dietary strategies for optimal dyslipidemia management. Interestingly, a clinical study has shown that a decrease in the amount of polyunsaturated fatty acids (PUFAs), and especially n-3 fatty acids, in membrane phospholipids may contribute to life span extension with calorie restriction [30]. Therefore, we should consider dietary fat composition with appropriate calorie restriction in regards to human health.

In this study, PS-matched comparisons of individuals with or without hLDL revealed that those with hLDL had higher consumption of SFA and lower intake of carbohydrates. These results are consistent with previous findings of a strong association between SFA intake and LDL-C [31,32]. High SFA intake has been associated with an increased risk of CVD, primarily mediated through increased levels of LDL-C [33]. However, several recent studies reported that some foods relatively rich in SFA, such as whole-fat dairy, dark chocolate, and unprocessed meat, are not associated with an increased risk of CVD, suggesting that different SFA have different biologic effects, which are modified by the food matrix and carbohydrate content of the diet [34]. Although the design of our study prevented evaluation of various dietary sources of SFA, we compared KHEI scores (which reflect the intake of culturally neutral basic food groups) to assess the quality of food intake, independent of food quantity [35]. Among the 14 KHEI components, the percentage of individuals with a high score for the “SFA” or “whole grains” components was significantly higher in the non-hLDL group than in the hLDL group. Consumption of plant-based foods has emerged as a promising approach to reduce LDL-C levels [36]. Whole grains, such as rice, corn, barley, and rye, are rich sources of fiber, which help lower LDL-C [37]. According to meta-analyses, whole-grain intake lowers serum TC and LDL levels [38], and ≥2 servings of whole grains per day are associated with a 10–20% reduced risk of developing CVD [39]. Therefore, current dietary guidelines recommend that total fat intake not exceed 30% of total caloric intake [40], and foods rich in healthy fats (e.g., fish, nuts, avocados, and seeds) be encouraged [41]. Replacing SFA with unsaturated fatty acids, proteins, or carbohydrates is also known to reduce LDL-C [31,32].

When comparing hTG and non-hTG groups, we found that the hTG group had a higher dietary Na/K ratio. Interestingly, this result was also observed when comparing the hLDL-only group with the hTG-only group. The hTG-only group had higher carbohydrate intake, as well as a higher dietary Na/K ratio than the hLDL-only group. Na and K are necessary for normal cellular function and have an inverse relationship. As dietary K can attenuate salt sensitivity and hypertensive effects of salt, the biological effects of Na and K intake should be considered jointly [42]. Some studies have suggested that the dietary Na/K ratio is more strongly associated with blood pressure and subsequent CVD than Na or K intake [42]. However, the role of a high Na/K ratio in lipid metabolism and its relationship to CVD are not well established.

For optimal health, the World Health Organization recommends restricting Na intake to 2 g per day, increasing K intake to 3.5 g per day, and maintaining a dietary Na/K ratio of approximately 1 [43]. The traditional South Korean diet is generally high in Na, with about half the population consuming >4000 mg/day of Na. This is far greater than the World Health Organization (WHO) recommendation [44]. The major dietary sources of K are fruits, legumes, whole grains, and vegetables. High salt intake and food processing decrease K use and eventually reduce K intake [45]. For this reason, we speculate that consumption of diets high in ultra-processed foods and “fast foods” with high-salt seasonings would increase the dietary Na/K ratio and have undesirable effects on TG levels. Conversely, intake of fresh fruits, fresh vegetables, and whole grains would reduce the dietary Na/K ratio and improve serum TG levels. The results of Mirmiran et al. support this speculation: fast food intake was positively associated with serum TG, and a higher dietary Na/K ratio increased the risk of hTG by 63% [46]. Our findings detecting an association between dietary Na/K ratio and hTG further highlight the importance of reducing Na and increasing K intake and provide a guide for potential interventional targets to decrease the risk of hTG.

High KHEI scores for “having breakfast” and “milk and dairy products” were more frequent in patients without hTG than in those with hTG. Skipping breakfast is considered an unhealthy habit, and several studies have shown its possible detrimental effects on lipid profiles [47]. A recent large population study in the United States demonstrated an increased risk of dyslipidemia among adults who skipped breakfast [48]. The association between skipping breakfast and dyslipidemia may be attributed to an increased appetite and subsequent high caloric intake after prolonged overnight fasting. Regarding milk or other dairy products, the effects of dairy products on CVD remain controversial because they are not only rich in SFA but also have high quantities of conjugated linoleic acid, whey protein, vitamins, and minerals, all of which have health benefits [49]. East Asians, and especially Koreans, tend to consume lower quantities of dairy products than Westerners [50]. A recent Korean study evaluating the risk of metabolic syndrome components according to milk intake detected protective effects of high milk consumption on TG and HDL-C levels. Frequent consumption of dairy products was also associated with a lower prevalence of hTG [50]. Therefore, reducing breakfast skipping and increasing dietary milk and other dairy products can be regarded as beneficial strategies for improving TG profiles in Koreans.

Our study has some limitations. Firstly, FFQ is less accurate than recording absolute nutrient intake values. Secondly, this is a cross-sectional study, which prevented us from assessing causality. Lastly, our results may not be generalizable to other races and ethnicities. Despite these weaknesses, this is the first study to investigate daily nutrient intake and dietary patterns in individuals with hLDL or hTG in a large Korean population-based study.

## 5. Conclusions

In conclusion, we found that hLDL was associated with high SFA intake, and hTG was associated with a high dietary Na/K ratio in the Korean population. The dietary Na/K ratio was a nutritional parameter that differentiated dietary habits between patients with only hLDL and those with only hTG. Furthermore, we found that reduced intake of whole grains was associated with hLDL, and skipping breakfast and low intake of milk and dairy products were associated with hTG. Our results may be used as a tool for monitoring patients with hLDL and hTG and for providing individualized dietary information and nutritional counseling. They may also aid in the future development of guidelines for dyslipidemia and CVD prevention, especially in the Korean population.

## Figures and Tables

**Figure 1 nutrients-13-03008-f001:**
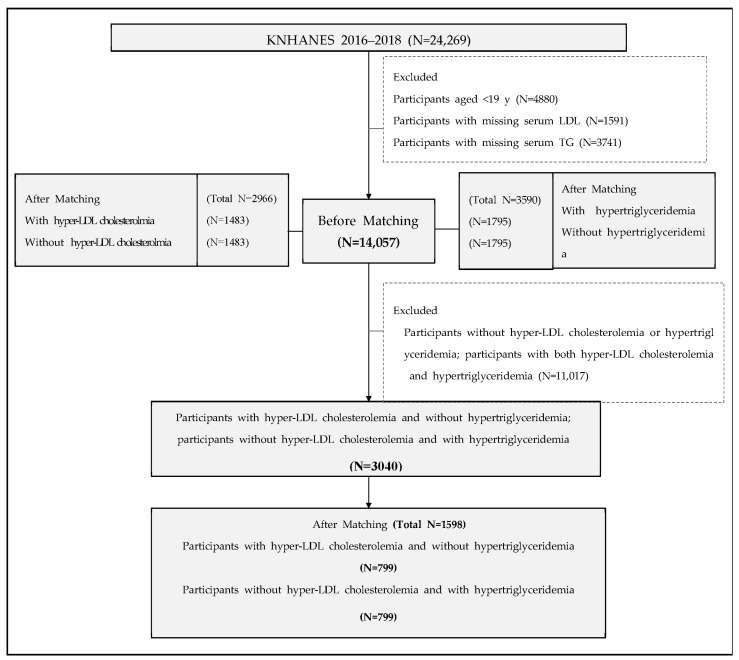
Study flow chart. KNHANES, Korea National Health and Nutrition Examination Survey; LDL, low-density lipoprotein.

**Figure 2 nutrients-13-03008-f002:**
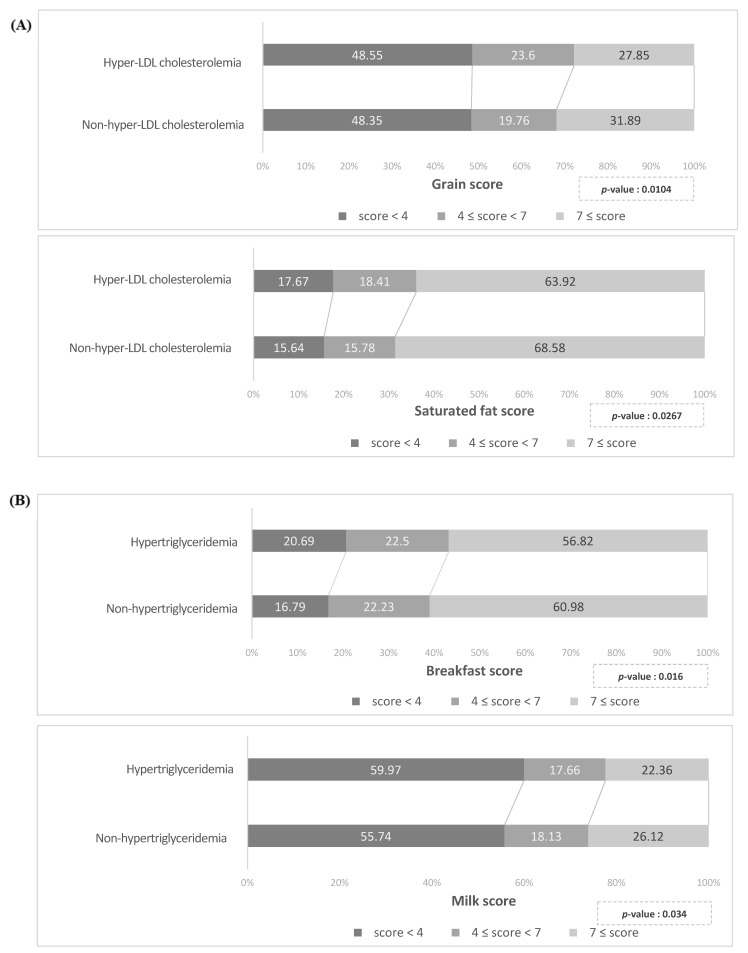
Percentages of participants in different categories for various Korean Healthy Eating Index components. Each Korean Healthy Eating Index component was categorized into three groups: low score: 0 ≤ score < 4, middle score: 4 ≤ score < 7, and high score: ≥7 points. *p*-values were calculated using the chi-squared test. (**A**) Percentages for each category of “whole grains” and “saturated fatty acids” scores in participants with or without hyper-LDL cholesterolemia. (**B**) Percentages for each category of “having breakfast” and “milk and dairy products” scores in participants with or without hypertriglyceridemia.

**Table 1 nutrients-13-03008-t001:** Clinical characteristics of study population according to the presence of hyper-low-density lipoprotein cholesterolemia.

	Before Propensity Score—Matching				After Propensity Score—Matching			
	Total (N = 14,057)	Non-hLDL (N = 12,745)	hLDL (N = 1312)	*p*-Value	Total (N = 2966)	Non-hLDL (N = 1483)	hLDL (N = 1483)	*p*-Value
Sex (male)	5955 (42.4)	5486 (43.0)	469 (35.8)	<0.001	1126 (37.96)	561 (37.83)	565 (38.10)	0.880
Age (y)	52.0 ± 16.6	51.7 ± 17.1	54.6 ± 14.3	<0.001	54.1 ± 15.2	54.1 ± 16.3	54.0 ± 14.0	0.816
BMI (kg/m^2^)	23.9 ± 3.5	23.9 ± 3.5	24.6 ± 3.4	<0.001	24.6 ± 3.5	24.6 ± 3.5	24.6 ± 3.5	0.720
Waist circumstance (cm)	82.4 ± 10.2	82.2 ± 10.3	84.0 ± 9.4	<0.001	83.9 ± 9.7	83.7 ± 9.9	84.1 ± 9.5	0.258
SBP (mm Hg)	119.0 ± 16.9	118.7 ± 16.8	122.1 ± 17.5	<0.001	120.5 ± 16.9	119.3 ± 16.3	121.8 ± 17.4	<0.001
DBP (mm Hg)	75.3 ± 10.1	75.0 ± 10.1	77.7 ± 10.2	<0.001	76.5 ± 10.0	75.1 ± 9.6	77.9 ± 10.3	<0.001
Glucose (mg/dL)	100.8 ± 23.0	100.7 ± 22.9	101.4 ± 24.5	0.367	100.6 ± 22.2	99.4 ± 19.1	101.8 ± 24.8	0.004
TC (mg/dL)	192.6 ± 37.7	185.8 ± 32.0	257.9 ± 24.4	<0.001	223.5 ± 45.0	188.6 ± 31.5	258.4 ± 24.9	<0.001
HDL-C (mg/dL)	51.2 ± 12.6	51.1 ± 12.7	52.4 ± 11.9	0.002	51.3 ± 12.0	50.3 ± 12.1	52.3 ± 11.7	<0.001
TG (mg/dL)	131.4 ± 92.7	131.1 ± 94.9	134.4 ± 68.1	0.115	133.5 ± 81.6	130.2 ± 90.2	136.8 ± 72.0	0.026
LDL-C (mg/dL)	115.1 ± 34.0	108.5 ± 28.0	178.7 ± 17.9	<0.001	145.5 ± 40.4	112.2 ± 26.9	178.7 ± 18.1	<0.001
WBC count (×10^3^/µL)	6.2 ± 1.8	6.2 ± 1.8	6.2 ± 1.8	0.379	6.2 ± 1.8	6.2 ± 1.7	6.3 ± 1.8	0.142
CRP	1.22 ± 2.00	1.20 ± 2.00	1.39 ± 2.04	0.001	1.33 ± 2.03	1.29 ± 2.03	1.37 ± 2.03	0.284
Uric acid	5.05 ± 1.37	5.03 ± 1.37	5.17 ± 1.39	0.001	5.11 ± 1.37	5.03 ± 1.35	5.20 ± 1.39	0.001
Current alcohol drinker	7411 (53.1)	6822 (53.9)	589 (45.1)	<0.001	1366 (46.1)	678 (45.7)	688 (46.4)	0.713
Current smoker	5089 (36.5)	4669 (36.9)	420 (32.2)	0.001	996 (33.6)	499 (33.7)	497 (33.5)	0.938
Physically active	5868 (43.3)	5363 (43.7)	505 (40.1)	0.014	1187 (40.0)	591 (39.9)	596 (40.2)	0.851
Comorbidity								
HTN	4608 (32.9)	4226 (33.2)	382 (29.2)	0.003	884 (29.8)	453 (30.6)	431 (29.1)	0.377
DM	1793 (12.8)	1693 (13.3)	100 (7.6)	<0.001	231 (7.8)	109 (7.4)	122 (8.2)	0.373

Data are mean ± standard deviation or number (percentage). LDL-C, low-density lipoprotein cholesterol; hLDL, hyper-LDL cholesterolemia; BMI, body mass index; SBP, systolic blood pressure; DBP, diastolic blood pressure; TC, total cholesterol; HDL-C, high-density lipoprotein cholesterol; TG, triglycerides; WBC, white blood cell; CRP, C-reactive protein; HTN, hypertension; DM, diabetes. *p*-values were calculated using the independent *t*-test or chi-squared test.

**Table 2 nutrients-13-03008-t002:** Daily nutrient intake of study population according to the presence of hyper-low-density lipoprotein cholesterolemia.

	Before Propensity Score—Matching				After Propensity Score—Matching			
	Total (N = 14,057)	Non-hLDL (N = 12,745)	hLDL (N = 1312)	*p*-Value	Total (N = 2966)	Non-hLDL (N = 1483)	hLDL (N = 1483)	*p*-Value
Total calories (kcal)	1809.5 ± 777.9	1813.6 ± 778.9	1770.0 ± 768.1	0.070	1869.9 ± 832.9	1883.4 ± 843.7	1856.0 ± 821.7	0.401
Carbohydrates (%)	66.0 ± 11.9	66.0 ± 11.9	66.1 ± 12.1	0.932	66.4 ± 11.7	66.9 ± 11.6	65.9 ± 11.8	0.021
Carbohydrates (g)	293.0 ± 123.6	293.6 ± 123.9	286.5 ± 120.6	0.062	294.2 ± 125.3	298.1 ± 128.2	290.1 ± 122.1	0.102
Proteins (%)	14.8 ± 4.5	14.8 ± 4.5	14.7 ± 4.5	0.449	14.7 ± 4.4	14.5 ± 4.3	14.8 ± 4.5	0.190
Proteins (g)	67.4 ± 37.8	67.7 ± 37.8	65.3 ± 37.1	0.045	66.6 ± 37.0	66.6 ± 36.5	66.6 ± 37.6	0.999
Fats (%)	19.2 ± 9.7	19.2 ± 9.7	19.3 ± 9.8	0.807	18.9 ± 9.5	18.5 ± 9.3	19.4 ± 9.6	0.025
Fats (g)	40.9 ± 32.9	40.9 ± 32.8	40.3 ± 33.4	0.534	39.9 ± 31.2	39.1 ± 30.1	40.7 ± 32.3	0.211
SFA (%)	6.1 ± 3.7	6.1 ± 3.7	6.2 ± 3.7	0.580	6.0 ± 3.6	5.9 ± 3.5	6.2 ± 3.6	0.013
MUFA (%)	6.0 ± 3.8	6.0 ± 3.8	6.0 ± 3.9	0.819	5.9 ± 3.7	5.8 ± 3.6	6.0 ± 3.8	0.163
PUFA (%)	5.2 ± 2.9	5.2 ± 2.9	5.2 ± 2.9	0.696	5.2 ± 2.8	5.1 ± 2.8	5.3 ± 2.9	0.111
PUFA/SFA	1.09 ± 0.71	1.09 ± 0.71	1.08 ± 0.72	0.563	1.10 ± 0.73	1.12 ± 0.74	1.08 ± 0.72	0.247
(PUFA + MUFA)/ SFA	2.12 ± 0.96	2.12 ± 0.96	2.08 ± 0.96	0.169	2.12 ± 0.97	2.15 ± 0.99	2.09 ± 0.96	0.106
N-3 PUFA (%)	0.90 ± 0.89	0.89 ± 0.88	0.96 ± 0.96	0.022	0.91 ± 0.86	0.89 ± 0.81	0.93 ± 0.90	0.191
N-6 PUFA (%)	4.30 ± 2.47	4.30 ± 2.47	4.27 ± 2.44	0.670	4.29 ± 2.45	4.22 ± 2.39	4.36 ± 2.50	0.145
N-6 PUFA N-3 PUFA	6.79 ± 4.87	6.81 ± 4.91	6.56 ± 4.45	0.062	6.75 ± 6.52	6.64 ± 4.95	6.86 ± 7.81	0.381
Cholesterol (mg)	218.7 ± 210.8	219.2 ± 210.0	213.4 ± 219.0	0.368	215.5 ± 212.7	214.2 ± 206.6	216.1 ± 218.8	0.751
Fiber (g)	25.3 ± 14.4	25.3 ± 14.0	25.30 ± 14.3	0.966	25.9 ± 14.9	26.1 ± 15.2	25.8 ± 14.5	0.646
Na (g)	3.3 ± 2.1	3.3 ± 2.1	3.1 ± 2.1	0.048	3.2 ± 2.1	3.2 ± 2.0	3.2 ± 2.1	0.735
K (g)	2.8 ± 1.4	2.8 ± 1.4	2.8 ± 1.4	0.694	2.8 ± 1.4	2.8 ± 1.4	2.9 ± 1.4	0.315
Na/K	1.25 ± 0.67	1.25 ± 0.67	1.19 ± 0.64	0.002	1.22 ± 0.65	1.2 ± 0.68	1.19 ± 0.62	0.050

Data are mean ± standard deviation. hLDL, hyper-LDL cholesterolemia; SFA, saturated fatty acids; MUFA, monounsaturated fatty acids; PUFA, polyunsaturated fatty acids; N-3, omega-3; N-6, omega-6; Na, sodium; K, potassium. *p*-values were calculated using the independent *t*-test.

**Table 3 nutrients-13-03008-t003:** Clinical characteristics of study population according to the presence of hypertriglyceridemia.

	Before Propensity Score—Matching				After Propensity Score—Matching			
	Total (N = 14,057)	Non-hTG (N = 12,015)	hTG (N = 2042)	*p*-Value	Total (N = 3590)	Non-hTG (N = 1795)	hTG (N = 1795)	*p*-Value
Sex (male)	5955 (42.4)	4713 (39.2)	1242 (60.8)	<0.001	2168 (60.4)	1107 (61.7)	1061 (59.1)	0.117
Age (y)	52.0 ± 16.9	51.7 ± 17.2	53.5 ± 14.9	<0.001	54.1 ± 15.7	54.4 ± 16.6	53.7 ± 14.7	0.138
BMI (kg/m^2^)	23.9 ± 3.5	23.7 ± 3.5	25.7 ± 3.4	<0.001	25.3 ± 3.2	25.2 ± 3.3	25.3 ± 3.1	0.214
Waist circumstance (cm)	82.4 ± 10.2	81.3 ± 10.1	88.5 ± 8.7	<0.001	87.1 ± 8.6	86.5 ± 9.1	87.7 ± 8.1	<0.001
SBP (mm Hg)	119.0 ± 16.9	118.2 ± 16.8	124.1 ± 16.0	<0.001	122.8 ± 16.4	121.8 ± 16.6	123.8 ± 16.1	0.002
DBP (mm Hg)	75.3 ± 10.1	74.6 ± 9.8	79.4 ± 10.8	<0.001	77.7 ± 10.4	76.3 ± 10.1	79.2 ± 10.5	<0.001
Glucose (mg/dL)	100.8 ± 23.0	99.3 ± 21.2	109.6 ± 30.4	<0.001	106.6 ± 27.5	105.0 ± 25.5	108.2 ± 29.1	0.004
TC (mg/dL)	192.6 ± 37.7	189.6 ± 36.4	210.3 ± 40.3	<0.001	199.5 ± 39.8	187.9 ± 36.1	211.2 ± 40.0	<0.001
HDL-C (mg/dL)	51.2 ± 12.6	52.7 ± 12.4	42.2 ± 9.4	<0.001	46.1 ± 11.2	50.0 ± 11.6	42.3 ± 9.3	<0.001
TG (mg/dL)	131.4 ± 92.7	103.0 ± 41.7	298.5 ± 127.6	<0.001	205.3 ± 132.6	112.0 ± 41.6	298.5 ± 126.6	<0.001
LDL-C (mg/dL)	115.1 ± 34.0	116.2 ± 32.8	108.5 ± 39.8	<0.001	112.3 ± 36.5	115.5 ± 32.8	109.2 ± 39.7	<0.001
WBC count (×10^3^/µL)	6.2 ± 1.8	6.1 ± 1.7	6.9 ± 1.8	<0.001	6.6 ± 1.7	6.4 ± 1.7	6.8 ± 1.8	<0.001
CRP	1.22 ± 2.0	1.19 ± 2.05	1.37 ± 1.74	<0.001	1.33 ± 1.90	1.34 ± 2.09	1.33 ± 1.69	0.891
Uric acid	5.05 ± 1.37	4.92 ± 1.31	5.77 ± 1.45	<0.001	5.52 ± 1.42	5.30 ± 1.35	5.74 ± 1.45	<0.001
Current alcohol drinker	7411 (53.1)	6189 (51.9)	1222 (60.3)	<0.001	2142 (59.7)	1080 (60.2)	1062 (59.2)	0.540
Current smoker	5089 (36.5)	3973 (33.3)	1116 (55.0)	<0.001	1921 (53.5)	977 (54.4)	944 (52.6)	0.270
Physically active	5868 (43.3)	5093 (44.0)	775 (39.7)	0.001	1444 (40.2)	726 (40.5)	718 (40.0)	0.785
Comorbidity								
HTN	4608 (32.9)	3693 (30.8)	915(44.9)	<0.001	1508 (42.0)	745 (41.5)	763 (42.5)	0.543
DM	1793 (12.8)	1365 (11.4)	428 (21.0)	<0.001	665 (18.5)	329 (18.3)	336 (18.7)	0.764

Data are mean ± standard deviation or number (percentage). hTG, hypertriglyceridemia; BMI, body mass index; SBP, systolic blood pressure; DBP, diastolic blood pressure; TC, total cholesterol; HDL-C, high-density lipoprotein cholesterol; TG, triglycerides; LDL-C, low-density lipoprotein cholesterol; WBC, white blood cell; CRP, C-reactive protein; Na, sodium; K, potassium; HTN, hypertension; DM, diabetes. *p*-values were calculated using the independent *t*-test or chi-squared test.

**Table 4 nutrients-13-03008-t004:** Daily nutrient intake of study population according to the presence of hypertriglyceridemia.

	Before Propensity Score—Matching				After Propensity Score—Matching			
	Total (N = 14,075)	Non-hTG (N = 12,015)	hTG (N = 2042)	*p*-Value	Total (N = 3590)	Non-hTG (N = 1795)	hTG (N = 1795)	*p*-Value
Total calories (kcal)	1809.5 ± 777.9	1799.0 ± 767.8	1873.6 ± 834.0	0.005	2025.6 ± 924.6	2010.2 ± 907.8	2041.5 ± 941.6	0.343
Carbohydrates (%)	66.0 ± 11.9	65.9 ± 11.9	66.7 ± 12.0	0.008	66.5 ± 11.9	66.4 ± 11.8	66.6 ± 12.0	0.640
Carbohydrates (g)	293.0 ± 123.6	290.9 ± 121.3	305.5 ± 136.5	<0.001	307.0 ± 132.2	307.4 ± 128.6	306.7 ± 135.9	0.892
Proteins (%)	14.8 ± 4.5	14.8 ± 4.5	14.8 ± 4.5	0.583	14.8 ± 4.4	14.9 ± 4.3	14.8 ± 4.5	0.704
Proteins (g)	67.4 ± 37.8	67.0 ± 37.4	70.4 ± 40.1	0.007	71.0 ± 39.6	71.0 ± 39.2	70.9 ± 40.0	0.905
Fats (%)	19.2 ± 9.7	19.3 ± 9.7	18.4 ± 9.9	0.004	18.7 ± 9.8	18.7 ± 9.7	18.6 ± 9.8	0.689
Fats (g)	40.9 ± 32.9	40.8 ± 32.6	41.1 ± 34.2	0.780	41.7 ± 33.9	41.9 ± 33.9	41.5 ± 34.0	0.734
SFA (%)	6.15 ± 3.71	6.20 ± 3.71	5.85 ± 3.73	0.003	5.92 ± 3.67	5.95 ± 3.58	5.89 ± 3.75	0.663
MUFA (%)	6.00 ± 3.80	6.05 ± 3.78	5.71 ± 3.91	0.006	5.80 ± 3.86	5.83 ± 3.81	5.76 ± 3.90	0.670
PUFA (%)	5.20 ± 2.86	5.23 ± 2.87	5.05 ± 2.84	0.018	5.13 ± 2.83	5.16 ± 2.83	5.09 ± 2.84	0.487
PUFA/SFA	1.09 ± 0.71	1.08 ± 0.71	1.14 ± 0.77	0.007	1.12 ± 0.74	1.11 ± 0.71	1.14 ± 0.76	0.213
(PUFA + MUFA)/SFA	2.12 ± 0.96	2.11 ± 0.95	2.16 ± 1.02	0.052	2.15 ± 0.98	2.13 ± 0.96	2.17 ± 1.01	0.246
N-3 PUFA (%)	0.90 ± 0.89	0.89 ± 0.88	0.91 ± 0.90	0.532	0.89 ± 0.82	0.88 ± 0.76	0.91 ± 0.87	0.286
N-6 PUFA (%)	4.30 ± 2.47	4.33 ± 2.48	4.13 ± 2.40	0.002	4.23 ± 2.43	4.28 ± 2.44	4.18 ± 2.41	0.245
N-6 PUFA N-3 PUFA	6.79 ± 4.87	6.82 ± 4.96	6.62 ± 4.33	0.076	6.63 ± 4.27	6.64 ± 4.26	6.62 ± 4.27	0.883
Cholesterol (mg)	218.7 ± 210.8	218.7 ± 208.9	218.5 ± 222.1	0.972	220.1 ± 220.6	220.6 ± 219.3	219.5 ± 222.1	0.889
Fiber (g)	25.3 ± 14.4	25.2 ± 14.4	26.1 ± 14.4	0.021	26.4 ± 14.4	26.6 ± 14.6	26.2 ± 14.3	0.491
Na (mg)	3.3 ± 2.1	3.2 ± 2.1	3.6 ± 2.3	<0.001	3.6 ± 2.3	3.5 ± 2.4	3.6 ± 2.3	0.316
K (mg)	2.8 ± 1.4	2.8 ± 1.4	2.9 ± 1.5	0.002	2.9 ± 1.5	3.0 ± 1.5	2.9 ± 1.4	0.338
Na/K	1.25 ± 0.67	1.23 ± 0.67	1.31 ± 0.67	<0.001	1.28 ± 0.67	1.26 ± 0.68	1.30 ± 0.66	0.049

Data are mean ± standard deviation. hTG, hypertriglyceridemia; SFA, saturated fatty acids; MUFA, monounsaturated fatty acids; PUFA, polyunsaturated fatty acids; N-3, omega-3; N-6, omega-6; Na, sodium; K, potassium. *p*-values were calculated using the independent *t*-test.

**Table 5 nutrients-13-03008-t005:** Daily nutrient intake of study population according to the presence of hyper-low-density lipoprotein cholesterolemia alone or hypertriglyceridemia alone.

	Before Propensity Score—Matching				After Propensity Score—Matching			
	Total (N = 3040)	hTG-Only (N = 1899)	hLDL-Only (N = 1141)	*p*-Value	Total (N = 1598)	hTG-Only (N = 799)	hLDL-Only (N = 799)	*p*-Value
Total calories (kcal)	1827.2 ± 805.0	1876.5 ± 833.9	1748.6 ± 750.3	<0.001	1906.6 ± 878.7	1938.3 ± 900.6	1875.4 ± 856.0	0.177
Carbohydrates (%)	66.3 ± 11.9	66.6 ± 11.9	65.9 ± 11.8	0.145	66.6 ± 11.8	67.8 ± 11.6	65.3 ± 11.9	<0.001
Carbohydrates (g)	296.9 ± 130.2	305.7 ± 136.9	282.8 ± 117.5	<0.001	296.0 ± 131.6	303.0 ± 139.8	289.0 ± 122.7	0.046
Proteins (%)	14.8 ± 4.5	14.9 ± 4.5	14.7 ± 4.5	0.469	14.7 ± 4.4	14.4 ± 4.1	15.0 ± 4.6	0.006
Proteins (g)	68.4 ± 38.1	70.7 ± 39.4	64.7 ± 35.5	<0.001	67.0 ± 36.8	66.3 ± 36.0	67.7 ± 37.5	0.487
Fats (%)	18.8 ± 9.7	18.5 ± 9.8	19.3 ± 9.6	0.034	18.7 ± 9.7	17.8 ± 9.6	19.6 ± 9.7	0.003
Fats (g)	40.7 ± 33.4	41.2 ± 33.9	39.8 ± 32.5	0.298	40.1 ± 33.5	38.3 ± 31.7	41.9 ± 35.1	0.041
SFA (%)	6.0 ± 3.7	5.9 ± 3.7	6.2 ± 3.6	0.021	6.0 ± 3.7	5.7 ± 3.7	6.3 ± 3.6	0.004
MUFA (%)	5.9 ± 3.9	5.8 ± 3.9	6.0 ± 3.9	0.083	5.9 ± 3.9	5.6 ± 3.8	6.2 ± 4.0	0.003
PUFA (%)	5.1 ± 2.8	5.1 ± 2.8	5.3 ± 2.9	0.085	5.1 ± 2.8	4.8 ± 2.7	5.4 ± 2.9	<0.001
PUFA/SFA	1.11 ± 0.74	1.13 ± 0.76	1.07 ± 0.70	0.037	1.11 ± 0.75	1.13 ± 0.77	1.09 ± 0.73	0.267
(PUFA + MUFA)/SFA	2.13 ± 0.99	2.16 ± 1.01	2.07 ± 0.95	0.032	2.14 ± 1.01	2.18 ± 1.04	2.11 ± 0.97	0.200
N-3 PUFA (%)	0.92 ± 0.89	0.89 ± 0.87	0.95 ± 0.93	0.117	0.91 ± 0.90	0.85 ± 0.79	0.97 ± 0.98	0.011
N-6 PUFA (%)	4.21 ± 2.42	4.15 ± 2.41	4.30 ± 2.44	0.136	4.18 ± 2.39	3.97 ± 2.30	4.38 ± 2.46	0.001
N-6 PUFA N-3 PUFA	6.64 ± 4.45	6.69 ± 4.39	6.57 ± 4.55	0.527	6.63 ± 4.55	6.60 ± 4.30	6.67 ± 4.79	0.782
Cholesterol (mg)	217.9 ± 220.3	220.7 ± 221.8	213.5 ± 217.8	0.414	212.6 ± 208.8	205.3 ± 208.4	219.8 ± 209.1	0.195
Fiber (g)	25.7 ± 14.3	26.0 ± 14.3	25.2 ± 14.2	0.147	26.0 ± 14.3	26.5 ± 14.7	25.4 ± 13.9	0.151
Na (mg)	3.4 ± 2.2	3.6 ± 2.3	3.1 ± 2.0	<0.001	3.3 ± 2.1	3.4 ± 2.1	3.2 ± 2.1	0.246
K (mg)	2.8 ± 1.4	2.9 ± 1.4	2.8 ± 1.3	0.082	2.8 ± 1.4	2.8 ± 1.4	2.8 ± 1.3	0.724
Na/K	1.26 ± 0.66	1.32 ± 0.67	1.18 ± 0.64	<0.001	1.22 ± 0.61	1.26 ± 0.63	1.19 ± 0.59	0.038

Data are mean ± standard deviation. LDL, low-density lipoprotein; hTG, hypertriglyceridemia; hLDL, hyper-LDL cholesterolemia; SFA, saturated fatty acids; MUFA, monounsaturated fatty acids; PUFA, polyunsaturated fatty acids; N-3, omega-3; N-6, omega-6; Na, sodium; K, potassium. *p*-values were calculated using the independent *t*-test.

## Data Availability

Data available in a publicly accessible repository; https://knhanes.kdca.go.kr/knhanes/main.do (accessed on 8 October 2020).

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
