# Peer review of "Differing Nutrient Intake and Dietary Patterns According to the Presence of Hyper-Low-Density Lipoprotein Cholesterolemia or Hypertriglyceridemia"

_nutrients, 2021, doi:10.3390/nu13093008_

Round 1
Reviewer 1 Report
By analyzing data from the KNHANES, the authors studied the correlation between nutrient intake (e.g. fat, carb, and breakfast) and CVD lipid risks with a PS matched statistical analysis. They show that high fat% intake was associated with high LDL-C levels and high carb% intake was associated with high TG levels. They also showed the correlation between missing breakfast and high TG levels, and the Na/K ratio in food intake in hLDL and hTG populations. This study is of interest to Nutrients’s readers.
Comments:
- In the section of 2.4 Assessment of nutrition intake and dietary patterns (2nd paragraph): It is not yet clear to show how the authors calculated the scores for Milk, breakfast, SFA, etc. More detailed information is required. For example, what does it mean for a participant who had a Milk score of 3? Also in Figure 2, the summary for percentages of score ≤ 3, 4 ≤ score ≤ 6 and 7 ≤ score, was 100%, does it mean that no participant’s score fell somewhere between 3 and 4, or between 6 and 7? If so, why?
- The history of nutrition intake (e.g. 3 months or 5 years) should be presented if possible because the information would be helpful to better study the correlation between nutrient intake and CVD risks.
- According to Table 2, the hLDL group has a higher fat% intake associated with a higher % SFA and PUFA intake. However, in Figure 2A the lower panel, the % for SFA score ³7 was lower in the hLDL group than the non-hLDL group (63.92% vs 68.58). How could the authors explain this paradox?
- In the Discussion (line 247-248): when the authors discussed the benefit of omega-3 intake for lipid profile, the total energy intake must be considered and discussed.
- I noticed that the Na intake in all groups in this study was higher than 2g/day. Thus the high Na intake in this study should be included In the Discussion.
Reviewer 2 Report
Undoubtedly high-quality article on a very prevalent topic of great importance for cardiovascular risk, due to the fact that high levels of LDL cholesterol and triglycerides are associated with a lower incidence of cardiovascular events.
It is a study with an adequate introduction, material and correctly planned method, highlighting a very high number of subjects included (24,269). The results are clearly expressed despite being very numerous and the discussion is adequate. A healthy diet is associated with lower levels of HDL cholesterol.
In summary, it seems to me a quality article that contributes a lot.
Author Response
I appreciate your careful review of our manuscript and the constructive comments.
It was a great honor to me to discuss this topic with you.